# Perceptual and Physiological Responses to Carbohydrate and Menthol Mouth-Swilling Solutions: A Repeated Measures Cross-Over Preliminary Trial

**Russ Best** [1,2,*], **Peter S. Maulder** [1,3] **and Nicolas Berger** [2]

1    Centre for Sports Science and Human Performance, Waikato institute of Technology,
     Hamilton 3288, New Zealand; Peter.Maulder@wintec.ac.nz
2    School of Health and Life Sciences, Teesside University, Middlesbrough TS1 3BX, UK; N.Berger@tees.ac.uk
3    Sports Performance Research Institute New Zealand, AUT University, Auckland 1010, New Zealand
*    Correspondence: Russell.Best@wintec.ac.nz

**Abstract:** Carbohydrate and menthol mouth-swilling have been used to enhance exercise performance in the heat. However, these strategies differ in mechanism and subjective experience. Participants (n = 12) sat for 60 min in hot conditions (35 °C; 15 ± 2%) following a 15 min control period, during which the participants undertook three 15 min testing blocks. A randomised swill (carbohydrate; menthol; water) was administered per testing block (one swill every three minutes within each block). Heart rate, tympanic temperature, thermal comfort, thermal sensation and thirst were recorded every three minutes. Data were analysed by ANOVA, with carbohydrate intake controlled for via ANCOVA. Small elevations in heart rate were observed after carbohydrate (ES: 0.22 ± 90% CI: −0.09–0.52) and water swilling (0.26; −0.04–0.54). Menthol showed small improvements in thermal comfort relative to carbohydrate (−0.33; −0.63–0.03) and water (−0.40; from −0.70 to −0.10), and induced moderate reductions in thermal sensation (−0.71; from −1.01 to −0.40 and −0.66; from −0.97 to −0.35, respectively). Menthol reduced thirst by a small to moderate extent. These effects persisted when controlling for dietary carbohydrate intake. Carbohydrate and water may elevate heart rate, whereas menthol elicits small improvements in thermal comfort, moderately improves thermal sensation and may mitigate thirst; these effects persist when dietary carbohydrate intake is controlled for.

**Keywords:** carbohydrate; menthol; thermal comfort; thermal sensation; thirst; water





## 1. Introduction

Mouth swilling is an increasingly popular ergogenic strategy employed by athletes over a short to moderate exercise duration [1–6], during nutrient restricted states [7–10] and may be appropriate during times of potential gastrointestinal distress [11,12]. Multiple nutritional stimuli are swilled, each conferring a different ergogenic effect [5,6,13,14] and magnitude thereof, most likely due to affecting differing sensory pathways. More precisely, the nutritional stimulus being swilled changes the cells targeted by and exposed to the swill, and the resultant ergogenic effect is the product of these interactions [15]. Nutritional stimuli that are swilled either directly or indirectly affect the brain and bypass the digestive system, so reducing energy intake, and the risk of gastrointestinal distress, which is frequently reported during prolonged endurance activity [11,12,16,17] when caffeine [18] and or carbohydrate are ingested [11,19,20].

Carbohydrate (CHO) is considered the gold standard ergogenic mouth swilling strategy, with a wealth of literature documenting its efficacy in contrasting environments [21], nutritional states [9] and sports [22–24]. Mechanistically, CHO is shown to activate areas of the brain that are associated with behavioural, cognitive and emotional responses [3], with areas associated with motivation and motor control also stimulated [25]. The activation of

these higher order and efferent regions of the brain, as supported by functional magnetic resonance imaging (fMRI), provide strong explanation(s) for CHO mouth swilling's ergogenic effects to date, but CHO is also shown to affect receptors within the oral cavity [26], as are caffeine [27–29] and menthol [13,30,31].

Menthol is considered a trigeminal afferent, stimulating the trigeminal nerve [32,33] and associated TRPM8 receptors [34–36]. The trigeminal network innervates the ophthalmic, mandibular and maxillary regions, with menthol and other cold stimuli particularly affecting the maxillary region due to its proximity to the nasal and oral cavities [37], stimuli have almost direct access to nerve endings due to the lack of squamous epithelia covering mucosa [37]. Indeed, it is the stimulation of this collection of nerves that is responsible for sphenopalatine ganglioneuralgia, or 'brain freeze' [38]. This potent response highlights the sensitivity and role within the cold temperature detection of TRPM8 receptors, and is likely enhanced due to the thinness of the membrane within the oral cavity [13,39]. Building the upon work performed by food scientists and psychologists [40,41], sports scientists have recently begun to investigate menthol mouth swilling as a strategy to ameliorate feelings of thermal comfort (TC) and sensation (TS) and exercise performance in hot conditions [13,42–45]. Menthol may also confer hedonic and thirst attenuating responses that may be beneficial to athletes, and potentially of use in other high-performing professions, e.g., firefighters or the military. These effects may in part be confounded by exercise due to effects such as increased ventilation [46,47] and decreased salivary flow rate [48]; but may be enhanced in hot conditions due to menthol's stimulatory effect upon TRPM8 receptors and the long-documented preference for the application of cold stimuli to the tongue and oral-cavity under thermally challenging circumstances [30,48,49].

The assessment of the effect of differing mouth swilling strategies on physiological and subjective measures, under resting conditions, may further elucidate mechanistic differences between nutritional stimuli applied to the oral cavity, without any confounding effects brought about by exercise. Therefore, the aim of this investigation was to quantify the physiological and subjective responses to CHO and menthol mouth swilling, at rest under thermally challenging conditions.

## 2. Materials and Methods

Ethical approval for this investigation was granted by the Teesside University School of Social Sciences, Business and Law ethics board.

### 2.1. Participants

Twelve participants (11 males and one post-menopausal female) took part in this investigation. Participants were recruited from an opportunity sample, with the sample size chosen so that each possible tasting order could be tested twice (e.g., solutions A, B, C). Participants had a mean age of 31.45 years (±90% CI: 26.88 to 36.02 years), and were 177.38 cm (172.99 to 181.76 cm) tall, weighing on average 75.87 kg (70.91 to 80.82 kg). Participants were made aware of the aim, procedure and risks of the study prior to providing informed written consent; participants were non-heat acclimated and were screened for medical issues that may have affected their ability to participate in the investigation prior to commencement. Participants wore sportswear appropriate for the environmental conditions, with an estimated insulative value of 0.45–0.61 clo [50,51] for male, and 0.57 clo for female participants, respectively.

### 2.2. Mouth Swilling Solutions

Solutions were prepared outside of the environmental chamber, under thermoneutral conditions (22 ± 0.5 °C) and administered in 25 mL aliquots. Five swills took place per swill condition; the swills lasted ~10 s prior to expectoration, with swilling order randomised via a Latin square design, using a customised spreadsheet [52].

Menthol (MEN) was prepared to a 0.1% (*v/w*) concentration, as per [46]. Briefly, a 5% menthol ethanol-based stock solution was diluted to the desired concentration using

distilled water. The carbohydrate mouth-swill (CHO) was prepared from unflavoured Maltodextrin (MyProtein, Northwich, United Kingdom), diluted to 10% (*v/w*) concentration (100 g·L$^{-1}$). Water acted as the placebo swill and a control period of no swilling was incorporated into each testing session (see Procedure). Quasi-single blinding was employed, whereby solutions were matched to be colourless, but were not matched for taste. Solutions were known to the experimenter who accompanied participants in the heat chamber, to ensure appropriate medical attention could be given, in the unlikely case of an adverse reaction [53].

*2.3. Procedure*

This investigation employed a repeated measures crossover design with the testing order of treatments assigned via Latin square; two participants completed each possible trial order. There were six possible trial orders. All testing took place within an environmental chamber set at 35 °C and 10% humidity, with outcome measures assessed at three minute intervals. This duration was chosen to allow sufficient observations to assess changes within and between conditions and has previously been deemed sensitive enough to assess alterations in TC and TS using the same scales as in the present investigation [54].

Testing began with 15 min of passive sitting, during which the time outcome data were recorded by the researcher, but no swilling took place. Following this control period (CON), participants commenced the swilling of solutions. Each solution was swilled five times at three-minute intervals, before progressing onto the next solution (note shaded areas in Figure 1). Once the final swill was completed and outcome measures recorded, the participants exited the chamber. The experimental procedure is pictorially represented in Figure 1.

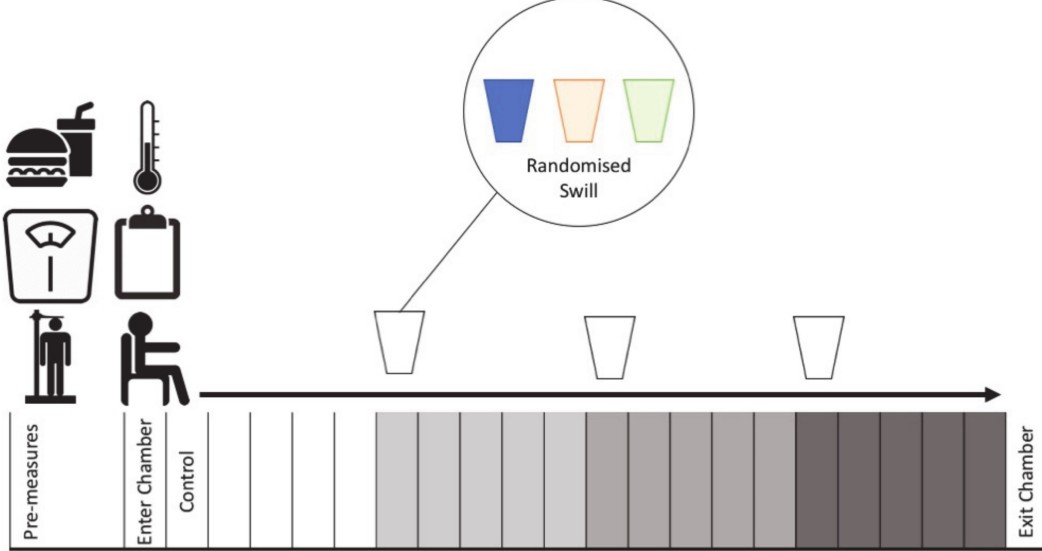

**Figure 1.** Experimental procedure: black vertical lines from entering the chamber represent three-minute intervals; at each interval, physiological and subjective measures were assessed as indicated by the clipboard and thermometer. Blank vessels represent the commencement of a new randomised mouth swill; this is supported with a change in shade of time interval cells. Randomised swills included water (blue), carbohydrate (orange) and menthol (green).

*2.4. Outcome Measures*

2.4.1. Physiological Measures

Tympanic temperature (T$_{tym}$) was assessed using a tympanic thermometer (±0.05 °C; Squirrel SQ10 Data Logger, Grant Instruments, Cambridge, United Kingdom), with measures taken from the ear contralateral to participants' dominant hand. Temperature was assessed prior to the administration of mouth swills, so any potential increase in temper-

ature caused by swilling or local irritation would be mitigated. Heart rate (HR) values were recorded 10 s prior to each three-minute interval via telemetry (Polar RS400; Polar, Helsinki, Finland).

### 2.4.2. Subjective Measures

Subjective measures were assessed using validated rating scales, with accompanying descriptors. Thirst was assessed via a 10-point scale [55], ranging from 'Not at all thirsty' to 'Extremely thirsty'. Zhang et al.'s scales of TC and TS were used to assess these qualities [54]. Both scales range from −4 to +4, with polar descriptors of very uncomfortable: very comfortable, and very cold: very hot, respectively. As a point of difference, the TC scale contains values of −0 and +0 to numerically describe just uncomfortable and just comfortable, respectively [54]. For the purposes of statistical analyses, these values are encoded as −0.5 and +0.5, to ensure distinction between the thermal states and congruity with the direction of participants' perceptions. To help differentiate between the perceptions of thermal comfort and sensation, participants were asked to imagine lying on a sun-lounger in a hot holiday destination. Thermal comfort was described as the degree of comfort experienced in such a circumstance, with factors such as fluid availability, clothing material and skin wettedness potentially influencing this perception. Thermal sensation was described as the degree of stress perceived as a result of environmental or metabolic heat sources, e.g., increased radiant heat load, windspeed or exercise. Participants acknowledged the difference between the two sensations and expressed readiness in their abilities to differentiate between the two characteristics.

### 2.4.3. Carbohydrate Intake

Carbohydrate intake (g) was calculated for each participant from a written 24 h food recall, using specialist software (Nutritics 2018, Nutritics, Dublin, Ireland). The 24 h food recall was administered by an appropriately trained sports nutritionist. These data were subsequently used as a covariate in the statistical analyses. A 24 h food recall was chosen, as opposed to a three or seven-day food diary, to reduce the burden placed upon participants. It is acknowledged that a more detailed method may be required to fully capture the variability and nature of participants' carbohydrate consumption; diaries/recall may be supported with food frequency questionnaires to compare against reported habitual intakes.

### 2.5. Statistical Analyses

Normality was assessed for using Skewness and Kurtosis tests (acceptable Z scores not exceeding +1 or −1). Initially, a two-way multiple analysis of variance (MANOVA) was conducted to determine the differences between time and beverage type on physiological and subjective outcome measures. Secondly, a two-way multiple analysis of covariance (MANCOVA) was conducted to determine the differences between the time and beverage type on outcome measures when controlling for carbohydrate intake. Significance was set at an a priori alpha level of $p < 0.05$. Effect sizes are reported as standardised mean differences ±90% confidence intervals (C.I.), with accompanying descriptors [56]. Descriptors accompanying effect sizes are demarked as follows: *Trivial*: 0–0.2; *Small:* 0.2–0.6; *Moderate*: 0.6–1.2; *Large:* 1.2–2.0; *Very Large*: >2.0. Ninety percent (90%) C.I. are used to differentiate between any observed significant results and the likely range in which true differences may occur [56,57], as opposed to another method of expressing a significant result.

## 3. Results

### 3.1. Carbohydrate Intake

The mean carbohydrate intake for participants was 69.92 g (±90% CI: 55.89 g to 83.94 g), with an absolute range of 203 g. These values are considered low in relation to participants' bodyweight [58], hence being stated in absolute as opposed to relative values.

*3.2. MANOVA*

There was a statistically significant interaction effect between the time and mouth-swill type on combined dependent variables, $F_{(20,750.507)} = 6.168$, $p < 0.0001$; Wilks' $\Lambda = 0.604$. This interaction effect is attributed to the significant effect of mouth-swill type on combined dependent variables, $F_{(10,452)} = 2.419$, $p = 0.008$; Wilks' $\Lambda = 0.901$, whereas time demonstrated a non-significant effect on combined dependent variables, $F_{(10,452)} = 1.090$, $p = 0.368$. Pairwise comparisons were used to identify significant effects upon dependent variables between mouth-swill types.

### 3.2.1. Physiological Outcomes

*Small* (ES: 0.26; from −0.04–0.54) significant differences in HR were observed between CON and water ($p = 0.018$). Small (0.22; from −0.09–0.52) non-significant differences in HR were also recorded between CON and CHO. All other HR comparisons were non-significant and *trivial*. Tympanic temperature during the CON period was significantly different to all other conditions (values; $p < 0.001$), displaying *moderate* effects (MEN: 0.89; 0.56 to 1.19. CHO: 0.91; 0.59 to 1.22. Water: 0.88; 0.56 to 1.19). Tympanic temperature displayed *trivial*, non-significant effects across all other comparisons, i.e., between swills.

### 3.2.2. Subjective Outcomes

Thermal comfort was significantly greater ($p < 0.002$) in CON compared to water swilling (*small*; −0.39; from −0.69 to −0.09). Despite not reaching statistical significance ($p < 0.062$) there were *small* (−0.32; from −0.63 to −0.02) differences in TC between CON and CHO conditions too, whereas MEN was only *trivially* different to CON (−0.01; −0.29–0.31). Menthol improved TC by a *small* magnitude compared to CHO (−0.33; from −0.63 to −0.03) and water (−0.40; from −0.70 to −0.10). Carbohydrate and water swilling were *trivially* different (0.10; −0.20 to 0.40) with respect to TC. Thermal sensation was *moderately* and significantly reduced by MEN in comparison to CON, CHO and water (see Table 1). All other comparisons were *trivially* and non-significantly different. Thirst was significantly greater in CON compared to MEN ($p < 0.001$) and water ($p < 0.011$), but not CHO ($p = 0.134$); the magnitudes of swilling's ability to improve thirst varied from *small* to *moderate* (see Table 1). Menthol lowered thirst significantly in comparison to CON and CHO, but not water; these differences were *moderate* in nature. Further contrasts are outlined in Table 1.

**Table 1.** Differences between mouth-swilling conditions for thermal sensation and thirst; significant effects are denoted by an asterisk (*). ES: effect size; C.I.: confidence interval.

| Variable | Swill | Comparison | *p* Value | ES; 90% C.I. | Descriptor |
|---|---|---|---|---|---|
| Thermal Sensation | Control | Menthol | 0.001 * | 0.66; 0.34 to 0.96 | *Moderate* |
| | | Carbohydrate | 0.835 | 0.04; −0.26 to 0.34 | *Trivial* |
| | | Water | 0.878 | 0.06; −0.24 to 0.36 | *Trivial* |
| | Menthol | Control | 0.001 * | −0.66; −0.96 to −0.34 | *Moderate* |
| | | Carbohydrate | 0.001 * | −0.71; −1.01 to −0.40 | *Moderate* |
| | | Water | 0.001 * | −0.66; −0.97 to −0.35 | *Moderate* |
| | Carbohydrate | Control | 0.835 | −0.04; −0.34 to 0.26 | *Trivial* |
| | | Menthol | 0.001 * | 0.71; 0.40 to 1.01 | *Moderate* |
| | | Water | 0.965 | 0.02; −0.28 to 0.33 | *Trivial* |
| | Water | Control | 0.878 | −0.06; −0.36 to 0.24 | *Trivial* |
| | | Menthol | 0.001 * | 0.66; 0.35 to 0.97 | *Moderate* |
| | | Carbohydrate | 0.965 | −0.02; −0.33 to 0.28 | *Trivial* |

**Table 1.** *Cont.*

| Variable | Swill | Comparison | $p$ Value | ES; 90% C.I. | Descriptor |
|---|---|---|---|---|---|
| Thirst | Control | Menthol | 0.001 * | 0.75; 0.43 to 1.06 | *Moderate* |
| | | Carbohydrate | 0.134 | 0.26; −0.05 to 0.56 | *Small* |
| | | Water | 0.011 * | 0.33; 0.02 to 0.63 | *Small* |
| | Menthol | Control | 0.001 * | −0.75; −1.06 to −0.43 | *Moderate* |
| | | Carbohydrate | 0.022 * | −0.55; −0.85 to −0.24 | *Small* |
| | | Water | 0.263 | −0.49; −0.79 to 0.18 | *Small* |
| | Carbohydrate | Control | 0.134 | 0.26; −0.56 to 0.05 | *Small* |
| | | Menthol | 0.022 * | 0.55; 0.24 to 0.85 | *Small* |
| | | Water | 0.259 | −0.07; −0.37 to 0.23 | *Trivial* |
| | Water | Control | 0.011 * | −0.33; −0.02 to 0.44 | *Small* |
| | | Menthol | 0.263 | 0.49; −0.18 to 0.79 | *Small* |
| | | Carbohydrate | 0.259 | 0.07; −0.23 to 0.37 | *Trivial* |

*3.3. MANCOVA*

Upon controlling for carbohydrate intake, there was a significant effect of mouth-swill type upon the combined dependent variables $F_{(10,298)} = 1.913$, $p < 0.043$; Wilks' $\Lambda = 0.883$. Between subjects' comparisons revealed significant differences for TS ($p < 0.004$) and thirst ($p < 0.048$). Heart rate ($p < 0.598$) and $T_{tym}$ ($p < 0.634$) responses were not significantly different between conditions when carbohydrate intake was controlled for, nor were differences in TC ($p < 0.151$).

Despite non-significant differences in TC ($p < 0.151$), when compared to both CHO (0.29; −0.09 to 0.65) and water (0.41; 0.04 to 0.78), MEN improved TC to a *small* extent. Pairwise comparisons demonstrated that MEN significantly reduced TS in comparison to CHO (−0.36 units; $p < 0.004$) and water (−0.37 units; $p < 0.008$), exerting *moderate* (−0.63; from −1.00 to −0.25) and *small* (−0.38; from −0.75 to −0.01) effects, respectively. Similar reductions in thirst were also observed, however, in contrast to the unadjusted model, MEN displayed a *moderate* (−0.67; from −1.04 to −0.29) standardised mean difference in thirst compared to water of −0.69 units ($p < 0.023$), with a *small* (−0.46; from −0.83 to −0.08) difference in comparison to CHO (−0.49 units; $p < 0.068$).

**4. Discussion**

The aim of this study was to assess the physiological and subjective responses to CHO and menthol mouth swilling, at rest under thermally challenging conditions, by employing a randomised tasting order in a quasi-blinded fashion. We acknowledge that due to a limited sample size and experimental design, these results are preliminary, but nevertheless, feel that they point to valuable future research directions.

Thermal sensation was significantly improved to a *moderate* degree by menthol in comparison to all other conditions. This finding has been reported repeatedly when menthol is applied to the oral cavity [42–45] and topically [59,60] by other researchers. However, we are the first group to document that this effect remains when nutrition (CHO intake) is statistically accounted for. This is important given the documented and potential use of menthol mouth swilling as an ergogenic aid during endurance exercise in thermally challenging conditions [13,44,61], where athletes may perform exercise across a range of states of carbohydrate availability, e.g., fasted, fed, or carbohydrate-loaded [62–64].

Furthermore, this suggests that menthol mouth swilling has the potential to be incorporated alongside other nutritional practices that may not alter TS, such as CHO intake during or following exercise. Such findings may be of use to athletes undertaking heat acclimation training, whereby the heat stimulus may be actively applied, i.e., during exercise [65–67] or passively via hot water immersion [68–70] or a sauna [71] during recovery from exercise. Alternatively, in competition, this finding allows athletes to pursue complementary nutritionally and thermally ergogenic strategies [72], potentially mitigating commonly reported issues during prolonged exercise (in the heat) such as gastrointesti-

nal distress [12,17,73] or taste fatigue [73]. This finding also has relevance to armed or emergency service personnel, who may have to report rapidly to situations in thermally challenging environments, potentially in varying states of nutritional preparedness.

Thirst, on the other hand, may be a key indicator of physiological readiness in these professions, and in prolonged endurance activity may also convey homeostatic information. Menthol mouth rinsing likely satiates thirst via a pre-absorptive pathway [50] through the stimulation of oral cold receptors [30,49], concomitantly conferring an hedonic effect, effectively mimicking a cold beverage. The hedonic relationship between beverage temperature is well described in humans [74–76], and has been shown to occur in rodents even in the absence of thirst or water deficit [77]. Therefore, when implementing mouth-swilling protocols, menthol's ability to significantly attenuate thirst, to a *small* to *moderate* extent, is something practitioners and scientists must consider. It is not clear from this investigation whether a brief application of menthol in a mouth-swill can alter exercise or thermoregulatory behaviours to the extent that they become detrimental to the individual in question. It would be prudent to recommend that menthol mouth-swilling be employed in compensable heat stress, in exercise durations whereby muscle glycogen concentration is not also a limiting factor so reducing the need for further nutritional support, e.g., events lasting ~60 min, or sports divided into periods of play. If athletes and practitioners still wish to employ menthol mouth-swilling in events outside of these constraints, then the co-implementation of other pre, or per-cooling strategies may be warranted [61,78,79] and these should be accompanied by the athlete or user education strategies from the supporting practitioner(s).

Thermal Comfort was also improved to a *small* extent by menthol mouth swilling when compared to CHO (−0.33; from −0.63 to −0.03) and water (−0.40; from −0.70 to −0.10). Conversely, menthol was *trivially* different to CON, displaying a broader confidence interval than for TS. Thermal comfort may be susceptible to a time effect in this investigation despite the randomised swilling order, as evidenced by participants reporting that CON was more thermally comfortable to a *small* extent, in comparison to CHO ($p < 0.062$) and water ($p < 0.002$) swills. As time progressed, participants may have experienced a greater awareness of the tactile elements of their environment such as the wettedness of clothing, local skin wettedness or the texture of the chair on which they were sat, as longer exposure to a hot environment elicits and accumulates a greater volume of sweat, through which a participant must interact with their tactile environment. Incorporating local measures of TC and skin wettedness in subsequent investigations would allow for greater precision in this hypothesis.

Similarly, the duration between swills warrants attention, as if swills are administered too close together, and receptors may still be saturated and thus sensations may be falsely heightened, thus committing a type 1 error. This is likely cause for concern when swilling menthol, however, it could also apply to other tastants such as caffeine, capsaicin or quinine [15] and sweet carbohydrates [1–7,15].

Heart rate and $T_{tym}$ were different over the course of the investigation. This is somewhat counterintuitive as typically we would expect a concomitant increase in both metrics over time, but not necessarily in response to swills. The expected time course response would be attributed to progressive heat load [65,80]; yet in the present investigation, each swill appeared to produce a different response, when compared to the CON period, both water and CHO elicited *small* increases in HR, whereas menthol did not.

With respect to menthol, our findings are in keeping with those of Shepherd and Peart [81], who rebutted the results of Meamarbashi and Rajabi, who asserted that 10 days of supplementation with a peppermint oil solution has a stimulatory effect [47] increasing maximal HR achieved during a maximal exercise test by 8%, and also increasing a complement of other exercise-associated variables. The *small* increases in HR observed during the water and CHO swilling appear counterintuitive from a sport and exercise scientist's perspective, as an increase in HR would confer a cost to the athlete, especially in a hot environment where factors such as increased sweat rate, the resulting dehydration

and increased skin blood flow already add to the thermal physiological strain experienced by the athlete [82]. However, at rest, when paired with a pleasant stimulus, these responses are perfectly normal [83,84]. Indeed, these responses have been noted to be goal directed [85,86], and increases in HR are associated with expectancy [83], a higher perceived reward value in healthy individuals [85,86]. Carbohydrate and water both confer hedonic responses by stimulating either receptors associated with fuel availability [6] or oral cold receptors [30], respectively, and may convey a homeostatically derived sense of reward; thus, an elevation in HR is probable. An alternative explanation for the elevation in HR in the present study is that of habituation [83]. Heart rate (HR) responses have been shown to be greater in response to an habituated 15.4% sucrose solution in comparison to water (control) or quinine solution (bitter); this response is consistent between exposures and independent of participant expectation [83]. Menthol mouth swilling on the other hand, may be too novel a stimulus for participants to be habituated to and subsequently elicit a HR response, but its ability to be of hedonic value in the current investigation is evidenced by improvements in TC and TS.

*Future Research Considerations and Directions*

- Habituation to menthol mouth swilling requires further investigation; a comparison between frequent and infrequent users of oral hygiene products may present logical starting populations.
- A larger sample size with homogenous sub-groups (male/female; ethnicities) would allow for a fuller exploration of the trends raised in the present investigation. Given the association between TRPM8 receptors and latitude [87], genomic or metabolomic sequencing may complement this work.
- Despite non-significant trends in $T_{tymp}$ across the investigation, the addition of core and skin temperatures measurements would facilitate a more thorough description of the heat storage experienced over the experimental time-frame. This is important if these findings are replicated in exercise, where convective and evaporative cooling may increase due to the performance of the velocities attained (e.g., equestrian sports [88]).
- Similarly, the use of measured temperatures as a covariate, alongside nutritional status (especially during exercise), would potentially elucidate physiological accompaniments or drivers for sensory thresholds and responses.

## 5. Conclusions

Menthol mouth swilling improves the perceptions of TC and TS, and satiates thirst compared to mouth swilling other solutions. Carbohydrate intake can alter the perceptual characteristics of other swills, and thus the nutritional state of those undertaking mouth swilling strategies is a key consideration for supporting practitioners and users. Swilling carbohydrate and water may lead to *small* elevations in HR; this may be an anticipatory hedonic response, and/or habituated. Menthol mouth swilling only *trivially* affects HR, but habituation to menthol mouth swilling warrants further exploration.

**Author Contributions:** Conceptualization, R.B. and N.B.; methodology, R.B.; formal analysis, R.B.; investigation, R.B.; resources, R.B. and P.S.M.; data curation, R.B.; writing—original draft preparation, R.B.; writing—review and editing, R.B., N.B. and P.S.M.; visualization, R.B.; supervision, N.B.; project administration, R.B.; funding acquisition, P.S.M. All authors have read and agreed to the published version of the manuscript.

**Funding:** This research received no external funding.

**Institutional Review Board Statement:** The study was conducted according to the guidelines of the Declaration of Helsinki, and approved by the Institutional Review Board (or Ethics Committee) of Teesside University (TSSSBL2015, May 2015).

**Informed Consent Statement:** Informed consent was obtained from all subjects involved in the study.

**Data Availability Statement:** There are no publicly stored datasets associated with this paper; data are available upon request, from the corresponding author.

**Conflicts of Interest:** The authors declare no conflict of interest.

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
