# Peer review of "Perceptual and Physiological Responses to Carbohydrate and Menthol Mouth-Swilling Solutions: A Repeated Measures Cross-Over Preliminary Trial"

_beverages, doi:10.3390/beverages7010009_

Round 1

Reviewer 1 Report

This article investigated the effects of carbohydrate and menthol mouth-swilling on psychophysiological responses related to thermal comfort. The experimental methodology seems to be appropriate, the results and discussion are clearly presented and well written. I think that the paper may be acceptable for publication in the present form (or after minor revision if authors can enhance the contents about the following).

・Is there any data that measures the skin temperature of the participants?

・Is the amount of participants' clothing (CLO) controlled?

・How did you set the precautions from the day before the experiment to the start of the experiment? (No alcohol or caffeine, etc.)

 Since your study treated internal stimulations and crossover design, I think that the above contents are not essential. But, such data or mention can provide a more quantitative assessment of thermal comfort and improve the reliability of analysis when physiological indices were analyzed.

Author Response

Reviewer 1

Thank you for your complimentary review, we really appreciate it, and are happy to expand in the areas you’ve identified for improvement and or provide further clarification for the reviewer’s benefit.

  1. We unfortunately did not measure skin temperature for this experiment, we had intended to measure brain blood flow in response to swilling, however these files became corrupted upon analysis hence their not being included in the present manuscript. We have expanded the text accordingly to identify this as a future direction.
  2. We recommended participants wear comfortable clothing, such as shorts and t-shirt that they would perform sport in. From the appropriate ISO guideline and Annex, we estimate the clo value of these outfits to be between 0.45 and 0.61 CLO for males, and 0.57 clo for the female participant. This is now included.
  3. Participants were asked to refrain from caffeine and alcohol, but continue with their habitual diet. Given the varied taste responses to other foodstuffs, we felt that continuation of an habitual diet was best for this study design, although we acknowledge this may lack the macronutrient control of a standardised diet.

Reviewer 2 Report

Reviewer's comment on Manuscript Number: beverages-1004750

The subject of the manuscript falls within the scope of Beverages but according to the done review it can be accepted after major amendments:

 1.      Materials and methods: why the group of participants is not consistent (only males, or females, or equally). What were the criteria of these participants choice? Why is it so small – no significant result will be obtained for such group. Besides what is the control group? Was there any, if yes, data should be compared. 2.      Why such concentrations were used? There should be different scenarios (time and concentrations)  applied and then compared with biological data. 3.      Authors mention that subjective measures were assessed using validated rating scales, with accompanying descriptors. Were these participants instructed or thought somehow to describe their sensations? It should be described. 4.      Such experiments would make sense if conducted on a larger scale with a control group.5.      The participants should have similar characteristics. 6.      I don’t like expression trivially different – what does it mean???When taking into account such small group the results are not giving statistically significant.   

The paper can be accepted after major revision, because currently paper is missing  control group and there are several major flaws that should be taken into account.

Author Response

Thank you for your comments, and suggestions. We have taken them on board and where possible tried to incorporate them into the manuscript.

  1. The participants were an opportunity sample; we appreciate that if the group was more homogenous then our findings may have been more consistent, leading to more (a greater number of) statistically significant results. Due to the nature of the repeated measures design, with each participant acting as their own control, we did not employ a control group. Such methodologies are common in supplementation studies, and are a more prudent way to manage participants and resources. In order to confer elements of control, we included a control period and a water swill within the trial, in an effort to distinguish between non-swilling and swilling controls. Again, the use of a control only group, is not necessarily realistic (i.e. lacks ecological transfer) and would result in greater research costs for relatively little information gained, beyond the present design.
  2. The present concentrations were chosen as we had previously investigated the preferred concentration for menthol mouth-rinsing (Best et al., 2018); this research was published in this journal. Similarly, 10% carbohydrate solutions are commonly employed in carbohydrate mouth rinsing work and are used by athletes in competitive scenarios. With a view to translating these findings to exercise protocols, we feel that when taken together this provides sufficient rationale for the use of the present concentrations
  3. Thank you for this comment, we have included comment to this effect in the methods section. This now reads ‘Subjective measures were assessed using validated rating scales, with accompanying descriptors. Thirst was assessed via a 10-point scale [51], ranging from ‘Not at all thirsty’ to ‘Extremely thirsty’. Zhang et al.’s scales of TC and TS were used to assess these qualities [52]. Both scales range from -4 to +4, with polar descriptors of Very Uncomfortable: Very Comfortable, and Very Cold: Very Hot, respectively. As a point of difference, the TC scale contains values of -0 and +0 to numerically describe just uncomfortable and just comfortable, respectively [52]. For the purposes of statistical analyses, these values are encoded as -0.5 and +0.5, to ensure distinction between thermal states and congruity with the direction of participants perceptions. To help differentiate between perceptions of thermal comfort and sensation, participants were asked to imagine lying on a sun-lounger in a hot holiday destination. Thermal comfort was described as the degree of comfort experienced in such a circumstance, with factors such as fluid availability, clothing material and skin wettedness potentially influencing this perception. Thermal sensation was described as the degree of stress perceived as a result of environmental or metabolic heat sources e.g. increased radiant heat load, windspeed or exercise. Participants acknowledged the difference between the two sensations and expressed readiness in their abilities to differentiate between the two characteristics.’
  4. Please refer to point 1. Whilst we do not disagree with the reviewer with respect to sample size, given the research design an absolute control group neither makes practical sense nor would be a useful allocation of research funding at this time. We have included sample size as a future research recommendation ‘A larger sample size with homogenous sub-groups (male/female; ethnicities) would allow for a fuller exploration of the trends raised in the present investigation. Given the association between TRPM8 receptors and latitude (ref), genomic or metabolomic sequencing may complement this work.’
  5. Please see above recommendation, to this effect
  6. Trivially different refers to an observed effect size that is less than small, thus the effect is trivial. This can be statistically significant or not, as we have used effect sizes and null-hypothesis significance testing in a complementary fashion. We have provided explicit threshold for effect sizes, and would note that the paper that describes these thresholds has been cited >5300 times. The expression is also used in medical literature, most notably in meta-analyses of medications; thus, we wish to include the term trivial as it is used frequently across a range of disciplines and within high levels of evidence.

Reviewer 3 Report

Dear Authors,

Firstly I would like to congratulate you on an attempt to discuss such an important topic and to present the findings however, I have several concerns and I have provided you with the number of comments. I have attempted to divide them in Major and Minor comments for your pursuit. I sincerely hope that this comments/suggestions assist in the improvements of the manuscript.

Major:

  1. Procedure section – can authors provide a better explanation of the procedures related to the swilling of solutions. The way it is written, it appears that all three conditions were tested at the same time with 3 minutes in between rather than 5 blocks of 3 minutes. Also, did the authors recorded a saturation effect of the Menthol (or other solutions). Taking menthol every 3 minutes can saturate the receptors providing false sensations. Is there enough evidence to support the 3 min increments as timing. Please clarify this.
  2. Did the authors controlled for the body temperature for all solutions being taken at the 45th minute? This can cause some sensory problems of the particular solutions. Also, the carry-over effect in between the testing time is another concern.
  3. Although the authors have reported a relatively low CHO intake, the sources and types of CHO can influence the results (30g from soda is rather different than 30g form pasta). This also reflects the overall CHO sweetness perception and consumption of other foods.
  4. Furthermore, authors will require to more justify the reasoning as to why the CHO measured at only one time point is generalized as the overall CHO intake of the participants. The 24hr recall taken at a single point does not represent adequate CHO intake of the participants. In addition, some athletes also perform CHO loading before the competition which in the presented study is not accounted for.
  5. Can authors provide some figures related to the relationships and correlations as this will definitely complement the findings.

Minor:

  1. Line 57-61. Reword the sentence and also other scientists are investigating the menthol mouth swills (food scientist, psychologists, etc…).
  2. Line 125 – who has administered 24hr food recall? If it was self administered who has monitored and what procedures were used for data collection.
  3. The reference list needs updating and following the specific format required for the journal.

Author Response

Reviewer 3

Thank you for the kind words regarding our efforts, we welcome the review and feel the comments made, and our attempts to address them, have strengthened the manuscript. We hope the amendments are to your satisfaction, and again appreciate the separation of Major and Minor revisions. Please find responses to your suggestions, below:

Major:

  1. Procedure section – can authors provide a better explanation of the procedures related to the swilling of solutions. The way it is written, it appears that all three conditions were tested at the same time with 3 minutes in between rather than 5 blocks of 3 minutes. Also, did the authors recorded a saturation effect of the Menthol (or other solutions). Taking menthol every 3 minutes can saturate the receptors providing false sensations. Is there enough evidence to support the 3 min increments as timing. Please clarify this.

Thank you for this recommendation, we hope the reworded version is clearer. This section now reads ‘This investigation employed a repeated measures crossover design with testing order of treatments assigned via Latin square; two participants completed each possible trial order. There were six possible trial orders. All testing took place within an environmental chamber set at 35ºc and 10% humidity, with outcome measures assessed at three minute intervals. This duration was chosen to allow sufficient observations to assess changes within and between conditions, and has previously been deemed sensitive enough to assess alterations in TC and TS using the same scales as in the present investigation (ref).

Testing began with 15 minutes of passive sitting, during which time outcome data were recorded by the researcher, but no swilling took place. Following this control period (CON), participants commenced swilling of solutions. Each solution was swilled five times at three minute intervals, before progressing onto the next solution (note shaded areas in Figure 1). Once the final swill was completed and outcome measures recorded, participants exited the chamber. The experimental procedure is pictorially represented in Figure 1.’

We have also provided further comment re receptor saturation and tasting intervals – ‘Similarly, the duration between swills warrants attention, as if swills are administered too close together, receptors may still be saturated and thus sensations may be falsely heightened, thus committing a type 1 error. This is likely cause for concern when swilling menthol, however could also apply to other tastants such as caffeine, capsaicin or quinine (ref) and sweet carbohydrates (refs).’

 Did the authors controlled for the body temperature for all solutions being taken at the 45th minute? This can cause some sensory problems of the particular solutions. Also, the carry-over effect in between the testing time is another concern.

Thank you for this comment, in the present investigation we did not control for this, but have recommended that in future investigations researchers consider using temperature as a covariate in their analyses to minimise its potentially confounding influence, and allow for deeper investigation of physiological thresholds and their interaction with perceptions.

  1. Although the authors have reported a relatively low CHO intake, the sources and types of CHO can influence the results (30g from soda is rather different than 30g form pasta). This also reflects the overall CHO sweetness perception and consumption of other foods.
  2. Furthermore, authors will require to more justify the reasoning as to why the CHO measured at only one time point is generalized as the overall CHO intake of the participants. The 24hr recall taken at a single point does not represent adequate CHO intake of the participants. In addition, some athletes also perform CHO loading before the competition which in the presented study is not accounted for.

Thank you for the above comments, we found them interesting and of value. We acknowledge that athletes may consume different foods, that will have differing resultant metabolic effects. We only performed a 24-hour recall, as we have found that this provides a balance between obtaining some nutritional information and error due to participant recall. We are also aware that participants give up their time to participate, and whilst we acknowledge that likely 3-7day food logs are more appropriate to obtain habitual measures of carbohydrate intake, we did not feel that this was appropriate to ask of our participants. We have included wording regarding states of nutritional preparedness, as this presents a valuable direction for future research. Thank you again, for these suggestions.

  1. Can authors provide some figures related to the relationships and correlations as this will definitely complement the findings.

Thank you for the suggestion, we have included a further figure to capture the relationships between time and swills, across outcome measures.

Minor:

  1. Line 57-61. Reword the sentence and also other scientists are investigating the menthol mouth swills (food scientist, psychologists, etc…).

Thank you for this comment, the sentence has been divided in two, and amended to read as follows: ‘Building upon work performed by food scientists and psychologists (refs), sports scientists have recently begun to investigate menthol mouth swilling as a strategy to ameliorate feelings of thermal comfort (TC) and sensation (TS) and exercise performance in hot conditions [13,41-44]. Menthol may also confer hedonic and thirst attenuating responses that may be beneficial to athletes, and potentially of use in other high-performing professions e.g. firefighters or the military.’

  1. Line 125 – who has administered 24hr food recall? If it was self administered who has monitored and what procedures were used for data collection.

The food recall was administered by a researcher with an MSc in Sports Nutrition and experience performing these recalls with athletes. Hence, we have updated this to read ‘The 24-h food recall was administered by an appropriately trained sports nutritionist.’

  1. The reference list needs updating and following the specific format required for the journal.

Thank you, we have amended as best we can and will await further support from the journal if required.

Round 2

Reviewer 2 Report

The manuscipt was improved. It can be published in present form.

Author Response

Thank you very much for recommending acceptance

Reviewer 3 Report

Dear Authors

I would like to congratulate you on adequately addressing my comments/suggestions. I believe that this manuscript is improved.

Minor comments

L95 - please add (v/v) after 0.1%

L96 - please add (v/w) after 10% concentration

L286 - please add appropriate reference at the end of the sentence '(refs)'

L287 - please replace word 'behave' with 'were different'

References - some minor formatting issues.

Kind Regards

Author Response

Thank you for the recommendations, all comments have been addressed.